# Can Zooplankton Add Value to Monitoring Water Quality? A Case Study of a Meso/Eutrophic Portuguese Reservoir



Ivo Pinto [1,2,3,4], Sandra Nogueira [2,4], Sara Rodrigues [2,4,*], Nuno Formigo [2,4] and Sara C. Antunes [2,4,*]

1   ICBAS, Instituto de Ciências Biomédicas de Abel Salazar, Universidade do Porto,
    Rua de Jorge Viterbo Ferreira 228, 4050-313 Porto, Portugal
2   CIMAR/CIIMAR, Centro Interdisciplinar de Investigação Marinha e Ambiental, Universidade do Porto,
    Terminal de Cruzeiros do Porto de Leixões, Avenida General Norton de Matos, S/N,
    4450-208 Matosinhos, Portugal
3   UMIB-ICBAS, Unidade Multidisciplinar de Investigação Biomédica—Instituto Ciências Biomédicas
    Abel Salazar, Universidade do Porto, Rua de Jorge Viterbo Ferreira 228, 4050-313 Porto, Portugal
4   Departamento de Biologia, Faculdade de Ciências, Universidade do Porto, Rua do Campo Alegre S/N,
    4169-007 Porto, Portugal
*   Correspondence: sara.rodrigues@fc.up.pt (S.R.); scantunes@fc.up.pt (S.C.A.)

**Abstract:** Despite the key role of zooplankton communities in regulating the water quality of lentic ecosystems, they are absent from the list of biological elements of the Water Framework Directive (WFD) for the assessment of ecological status. Based on this, the present work was a case study that aimed to understand the relevance of zooplankton as a bioindicator for classifying the water quality of reservoirs. For one year and in each season, the water quality of the Torrão reservoir was assessed using the mandatory elements proposed by the WFD in the sampling year (second cycle) and the legislation currently applicable (third cycle). Additionally, zooplankton samples were collected to characterize the community dynamic. The water quality of the Torrão reservoir does not reach the WFD good ecological potential. Furthermore, with the updating of the criteria, the classification tends to get worse. Concerning the zooplankton, the occurrence of Cyclopoida and *Bosmina* are associated with lower water quality, as well as the collapse or low abundance of the *Daphnia* population, in the same periods. Low abundances of zooplankton are associated with better water quality, and the Shannon-Wiener diversity values decrease with the increase of the trophic state. High-efficiency feed filters and macrofiltrator organisms dominate the Torrão reservoir in all seasons, which is associated with low water quality. The Calanoida/Cyclopoida ratio shows a strong and negative association with the trophic state. The occurrence and abundance of species, Shannon-Wiener diversity, functional groups (high and low efficiency feed filters and macrofiltrators), and different ratios (large Cladocera/total Cladocera and Calanoida/Cyclopoida) are promising and valuable tools to determine the water quality status, and should be considered within the WFD metrics. Notwithstanding this, further research including reservoirs from different geographical areas with various trophic status and pressures must be analyzed.

**Keywords:** reservoirs; Water Framework Directive; trophic state; bioindicators; functional groups; zooplankton index; zooplankton ratios





## 1. Introduction

Water is an extremely important natural resource for living beings, ecosystems, and humans [1]. For a long time, water was considered an infinite resource; however, its use for all activities is on the threshold of exceeding the natural rate of water production on the planet [2]. Most problems related to water quality originate from intensive agriculture, industrial production, domestic and urban waste, and untreated wastewater [3]. Excessive nitrogen and phosphate loads, largely resulting from fertilizer application, are among the most common chemical contaminants recorded in the world's freshwater resources [3,4].

On the other hand, emerging compounds and pollutants (e.g., pesticides and antibiotics) have been detected in various ecosystems, contributing to the further degrading of water quality [5]. To minimize the impacts on freshwater aquatic ecosystems, it is important to understand the structure, composition, and dynamics of biotic communities in order to safeguard the ecological quality of these ecosystems [6].

Reservoirs are similar to natural lakes in terms of water storage capacity, low flow velocity, and abiotic characteristics in comparison with rivers [7]. However, natural lakes differ from reservoirs in their slow formation by natural geomorphological processes or rapid creation as a result of catastrophic events; processes of stratification; water mass circulation and hydrodynamics, controlled by natural rhythms; and biodiversity, which was historically selected and adapted to natural lake dynamics [8]. In addition, reservoirs show a faster increase in nutrient concentration (e.g., phosphorus and nitrogen) from the leachates of soils and adjacent vegetation [9,10], enhanced by anthropogenic activities (e.g., agriculture and industry). Furthermore, this water degradation can cause the abnormal growth of primary producers (promoting eutrophication processes), which can compromise the balance of the ecosystem [11,12].

The European Union approved the Water Framework Directive 2000/60/EC (WFD) in 2000, with the aim of standardizing the programs for the monitoring and sustainable management of waterbodies [13]. Preventing the deterioration and ensuring the protection and improvement of aquatic environments are important goals to be met under this Directive [14]. Within the WFD implementation, a typology was established for heavily modified water bodies, such as reservoirs. According to the WFD, in this case, the concept of ecological potential was applied, representing the deviation of the actual quality of the aquatic ecosystem from the maximum it can reach.

The WFD proposes phytoplankton as a biological element for reservoir characterization and water quality evaluation. The composition of the phytoplankton community is directly related to the trophic state. Thus, the variation of biomass and the time sequence of phytoplankton populations (seasonal succession) is related to abiotic components (e.g., nutrients) and biotic interactions (e.g., zooplankton and ichthyofauna) [15]. However, zooplankton, which plays a key role in the food web as a primary consumer, is not included in the assessment of the water quality of reservoirs [14,16,17]. In addition, this biological component has already been shown to be sensitive to chemical and physical variations in aquatic ecosystems, such as reservoirs [14]. Recognizing this ecological relevance, several authors have demonstrated that zooplankton should be incorporated into the WFD as a biological element to assess the ecological potential of lentic freshwater ecosystems [16–18].

Zooplankton is a group of heterotrophic organisms that occur in the water column, and their diet is not very diversified, since most are filter feeders feeding on seston, namely phytoplankton, bacteria, and organic detritus [15]. The structure and abundance of zooplanktonic communities, and their spatial distribution, can be influenced by abiotic factors (e.g., temperature, pH) as well as by interactions between species (e.g., quantity and quality of algae, bacteria, and fish) [19]. Zooplankton exhibits a diversity of ecological strategies and patterns of dominance. According to [20], the ecological strategies of zooplankton can be characterized according to the ecological function (feeding, survival, reproduction, and growth) and morphological, physiological, behavioral, and life cycle characteristics. These organisms are extremely sensitive to various environmental stresses (e.g., pesticides, metals, and drugs) [8,17,20–26], being affected in different ways (e.g., mortality and resistant egg production). Furthermore, an ecological functional approach has also been applied to zooplanktonic communities concerning the trophic state of lakes, thus explaining the ecological basis for community changes with eutrophication processes [19,21,27,28]. In addition, several authors have found that specific organisms are associated with the different trophic states of the water body, Cyclopoida, and some cladocerans (e.g., *Bosmina*) with eutrophic ecosystems, while Calanoida and other cladocerans (e.g., *Daphnia*) are associated with oligotrophic ecosystems [29–31].

The main objective of this study was to evaluate the relevance of the zooplankton community as a complementary bioindicator for the evaluation of the water quality of the Torrão reservoir as a case study. To this end, the Portuguese methodologies mandatory for the sampling year (the second cycle of the WFD) and the legislation currently applicable (the third cycle of the WFD) were used to assess the water quality of this reservoir and to determine whether the inclusion of the zooplanktonic communities provided relevant information about the ecological status of the water body.

## 2. Materials and Methods

### 2.1. Study Area

The Torrão reservoir is located in Marco de Canaveses city, Porto, Portugal, and is located in the Tâmega River, the longest and largest tributary of the Douro River in Portuguese territory [32]. From its source in Galicia, the Tâmega River runs for approximately 150 km, crossing the northeast border of Portugal, until it flows into the Douro River. The Torrão reservoir has a total capacity of over 120,000 dam$^3$, an operational capacity of 77,090 dam$^3$ (for hydroelectric production), and a full storage level quota of 65 m [33]. According to the WFD, this reservoir is classified as a north reservoir (a cold water reservoir located in the northern region in mountainous areas). The Torrão hydrographic basin is characterized by a very marked seasonality (hot, dry summers and cold winters), with characteristics of temperate climate transition zones. The annual temperature average is 15.2 °C, with an average of 21.2 °C in summer and 9.6 °C in winter. The average monthly precipitation in the Douro hydrographic basin (including the Torrão hydrographic basin) is approximately 83 mm, with a maximum in December, at 140 mm, and a minimum in July and August, with 17 mm [34]. The proximity to the Atlantic Ocean and the Mediterranean Sea strongly influences the climate of mainland Portugal [35]. The Mediterranean influence is felt mainly in the summer in the south and east of the territory, causing high temperatures and low rainfall. The Atlantic influence is felt primarily in winter and in the northwest of the country. It is responsible for high precipitation and the attenuation of the effects of dry and cold winds from the Iberian Peninsula's interior [35].

To conduct the present study, five sampling sites were defined (T1 to T5) on the banks of the Torrão reservoir (Figure 1), and each location was selected based on the heterogeneity of the surrounding landscape, pollution sources, and accessibility.

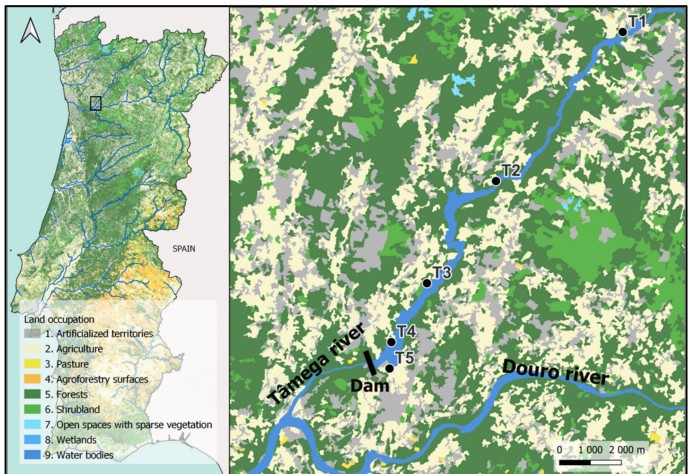

**Figure 1.** Geographic location of Torrão reservoir (black rectangle on the left map, zoom on the right), marked with the sampling sites: T1 (41°11′38.05″ N, 8°09′51.61″ W), T2 (41°09′01.59″ N, 8°12′47.31″ W), T3 (41°07′14.31″ N, 8°14′22.89″ W), T4 (41°06′12.48″ N, 8°15′12.12″ W) and T5 (41°05′44.77″ N, 8°15′14.63″ W). The different colors represent the first level of detail of land occupation according to the land use report (2018).

## 2.2. Sampling Procedure

All of the samples were collected from the margins, and the sampling was conducted seasonally (the summer and autumn of 2017, and the winter and spring of 2018). In situ, some physical and chemical parameters were measured on the water column using a multiparametric probe (a Multi 350i) [dissolved oxygen (% and mg/L), pH, conductivity (μS/cm), and temperature (°C)]. In addition, 5 L of water was collected and transported to the laboratory (in a thermal bag at 4 °C in the dark) for further analysis of other physical and chemical parameters (nitrites, nitrates, ammonia, phosphates, suspended solids, dissolved solids, turbidity, dissolved organic carbon, and biochemical oxygen demand) and biological elements (phytoplankton).

Zooplankton samples were collected from the margin using a 250 μm mesh hand net, and five hauls were performed at each site in order to ensure the same sampling effort. The samples were immediately stored and preserved in 70% alcohol for subsequent analysis (identification and counting) in the laboratory.

## 2.3. Physical and Chemical Analysis

In the laboratory, on the same day of the field trip, or no later than 24 h after sample collection, the other physical and chemical parameters of the water samples were measured [7]. Biochemical oxygen demand ($BOD_5$) was determined according to [36], the dissolved organic carbon (DOC) was determined indirectly through the color of the water (CDOC—colored dissolved organic carbon) according to the methodology established by [37], the turbidity was measured as described in [36], and the content of total suspended solids (TSS) and volatile suspended solids (VSS) were determined according to [38].

For nutrient quantification in each water sample, total phosphorus was quantified according to the methodology described in [39]. A benchtop photometer (Spectroquant Multy Colimeter) and specific test procedures were used to quantify nitrates ($NO_3^-$, test procedure 1.1477), ammonia ($NH_4^+$, test procedure 1.14752), and nitrites ($NO_2^-$, test procedure 1.4776).

## 2.4. Biological Analysis

The phytoplankton characterization was conducted using 800 mL of water sample from each sampling site, according to legal recommendations [40] (for more information see [12]). The final classification of biological elements is expressed as the ecological quality ratio (EQR), which results from the normalization of the four metrics (Cyanobacteria biovolume, algae group index (AGI), chlorophyll *a* concentration, and total biovolume) following the guidelines provided for the second and third cycles of WFD.

For macrozooplankton characterization, the samples were counted in their entirety using a binocular stereoscope. The identification of organisms in the Copepoda subclass was up to order due to the functional groups used for the index calculation proposed by Geller and Müller [41]. The identification of Cladocera was carried out on the species level whenever possible. The identification was aided by several identification guides [42–44]. Although freshwater ecosystems are composed of elements from different groups of invertebrates: protozoa, rotifers, Copepoda, and Cladocera, in the present study, only Copepoda and Cladocera were considered in community analysis. Nauplii, rotifers, and the protozoa were not quantified in this study since the trawl used in their collection was 250 μm, which is insufficient to capture most of these individuals.

## 2.5. Statistical Analysis

To comply with the WFD metrics for the assessment of the water quality of the reservoir, physical and chemical parameters were interpreted following the ecological quality standards (EQS) of the second [45] and third cycles of WFD [46] (Table S1). The biological element was interpreted using the ecological quality ratios (EQR) of the second [45] and third cycles of WFD [46] (Table S2).

A principal component analysis (PCA) was conducted to extract the main gradients underlying sites and seasons, using the software CANOCO 4.5®. Before the PCA analysis, the environmental variables were standardised, and the redundant variables were removed from the analysis.

The macrozooplankton analysis always used the same matrix data concerning the Geller and Müllers [41] species/group classification in order to interpret the results in terms of water quality. The abundance was determined by counting the total number of individuals in each *taxon*. The Shannon-Wiener index and Pielou's index were calculated to measure the diversity and evenness of each sample, respectively. The Large Cladocera/Total Cladocera and the Calanoida/Cyclopoida ratios were also calculated to discriminate the relevance of large Cladocera (which has a higher efficiency in controlling phytoplankton) and Calanoida (which is associated with more oligotrophic ecosystems) in the zooplankton community. The zooplankton was also analyzed using a functional approach, being classified according to the feeding strategies (low efficiency, high efficiency, and macrofiltrators) based on the type of filtration apparatus, namely the filter mesh-sizes and their implications on food selectivity (the ability to feed on bacteria and phytoplankton), as defined by [41]. Moreover, the association between relevant pairs of variables (environmental variables and biological indicators) was assessed using correlational analysis. The level of significance was set at 0.05, and the analysis was carried out with the custom function "rquery.cormat" [47] and the package "corrplot" [48] in R (version 4.1.3 [49]) using IDE RStudio (version 2022.2.1.461 [50]).

## 3. Results

### 3.1. WFD Ecological Potential

The results of the general physical and chemical parameters are shown in Table S1, and the biological element (phytoplankton) is shown in Table S2. The final ecological potentials are presented in Table 1 for the second and third cycles of WFD. The results obtained regarding the threshold values for the second cycle of the WFD revealed that the Torrão reservoir presents a good ecological potential in spring and winter (except for T3 in winter due to the increase of [$P_{total}$]), and moderate ecological potential in the remaining sampling sites and dates. The lower observed ecological potential classification is due to [$P_{total}$] (>0.05 mg/L; namely in autumn), pH (up to 9 in summer), and $O_2$ (<60% in autumn for T2 or >120%, in summer for all sampling sites) (Table S1). When the results were analyzed using the threshold values for the third cycle of the WFD, all stations present moderate ecological potential (Table 1). The change in the threshold values (e.g., $NO_3^-$ in the second cycle was 25 mg/L, and in the third cycle it was 3 mg/L), as well as the existence of more parameters with defined reference values in the third cycle (e.g., $NO_2^-$), could explain the lower ranking obtained compared to the ranking done with the threshold values of the second cycle of the WFD. Regarding the threshold values of the third cycle, only $NH_4^+$, $BOD_5$, and TSS values met the guidelines for good or excellent ecological potential for all sites and season samples. The sampling carried out in winter presented the best classifications due to lower Chl *a* concentration and total biovolume, as well as the absence of cyanobacteria and organisms associated with polluted waters (Table S2). It was possible to observe an increase in the Chl *a* concentration (an indirect measure of organisms abundance) at the beginning of spring, reaching a peak in the summer, then being gradually replaced by a species of Cyanobacteria, with a peak in autumn, and the subsequent death and degradation in winter (Table S2). Consequently, an apparent seasonality can be observed, where the worst water quality was recorded in spring, summer, and autumn (due to an increase of Chl *a* concentration and the total biovolume), independently of the sampling site.

**Table 1.** Ecological potential is achieved according to the second [45] and third [46] cycles of the WFD. The colors (green, yellow, and orange) represent the classification according to the WFD.

| | | Second Cycle of WFD | | | Third Cycle of WFD | | |
|---|---|---|---|---|---|---|---|
| | | Physical and Chemical Classification | Biological Classification | Final Classification | Physical and Chemical Classification | Biological Classification | Final Classification |
| T1 | Summer | Moderate | Moderate | Moderate | Moderate | Poor | Poor |
| | Autumn | Moderate | Moderate | Moderate | Moderate | Poor | Poor |
| | Winter | Good | Good or more | Good | Moderate | Moderate | Moderate |
| | Spring | Good | Good or more | Good | Moderate | Good | Moderate |
| T2 | Summer | Moderate | Moderate | Moderate | Moderate | Poor | Poor |
| | Autumn | Moderate | Good or more | Moderate | Moderate | Moderate | Moderate |
| | Winter | Good | Good or more | Good | Moderate | Good | Moderate |
| | Spring | Good | Moderate | Moderate | Moderate | Poor | Poor |
| T3 | Summer | Moderate | Moderate | Moderate | Moderate | Moderate | Moderate |
| | Autumn | Moderate | Good or more | Moderate | Moderate | Moderate | Moderate |
| | Winter | Moderate | Good or more | Moderate | Moderate | Good | Moderate |
| | Spring | Good | Poor | Poor | Moderate | Poor | Poor |
| T4 | Summer | Moderate | Good or more | Moderate | Moderate | Moderate | Moderate |
| | Autumn | Moderate | Good or more | Moderate | Moderate | Moderate | Moderate |
| | Winter | Good | Good or more | Good | Moderate | Excellent | Moderate |
| | Spring | Good | Poor | Poor | Moderate | Poor | Poor |
| T5 | Summer | Moderate | Good or more | Moderate | Moderate | Moderate | Moderate |
| | Autumn | Moderate | Moderate | Moderate | Moderate | Moderate | Moderate |
| | Winter | Good | Good or more | Good | Moderate | Good | Moderate |
| | Spring | Good | Moderate | Moderate | Moderate | Poor | Poor |

Based on the PCA applied to the physical, chemical, and biological parameters (Figure 2), a seasonality pattern was observed in the Torrão reservoir, with the sampling sites grouped by season. Summer was associated with high temperatures and pH, as well as with an increase of chlorophyll *a* and total biovolume. In autumn, the sites presented an increase in nutrient concentrations ($NH_4^+$ and $P_{total}$), as well as in Cyanobacteria biovolume and AGI values. In winter, an association between VSS and $NO_3^-$ was observed, while in spring, the sites were associated with higher concentrations of chlorophyll *a*. In addition, the physical and chemical parameters that support biological elements also reflected the typical dynamic of a phytoplankton community in a eutrophic reservoir.

### 3.2. Zooplankton Community

The zooplankton dynamics for each sampling site are presented in Table 2. Winter showed an abrupt decrease in zooplankton abundance, with only 50 organisms in T5. Overall, the highest abundances were recorded in spring and summer, mainly due to Copepoda (both Cyclopoida and Calanoida), *Chydorus*, *Ceriodaphnia*, and *Bosmina*. In T1, a decrease in the abundance of the Cyclopoida was observed in winter, with an increase in the abundance of small Cladocera species (*Chydorus* and *Alona*). In T2, an increase in Cyclopoida and *Bosmina* abundance was observed in spring and *Ceriodaphnia* in autumn. Site T3 was mainly characterized by the dominance of *Sida* (>90%) in the autumn and Cyclopoida in the spring. *Sida* is a littoral species living among macrophytes. Despite no macrophytes being recorded in the study, submerged herbaceous vegetation was observed in site 3 depending on the water level of the reservoir (e.g., autumn). Sites T4 and T5 showed a dominance of Calanoida (>50%) and *Ceriodaphnia* (>20%) in summer, followed by an increase in *Sida* and a decrease in Calanoida in autumn. Despite the low abundances recorded in winter, Cyclopoida (>50%) was the most abundant in both sites, with *Bosmina* (~40%) and *Chydorus* (>20%) also present in T4 and T5, respectively.

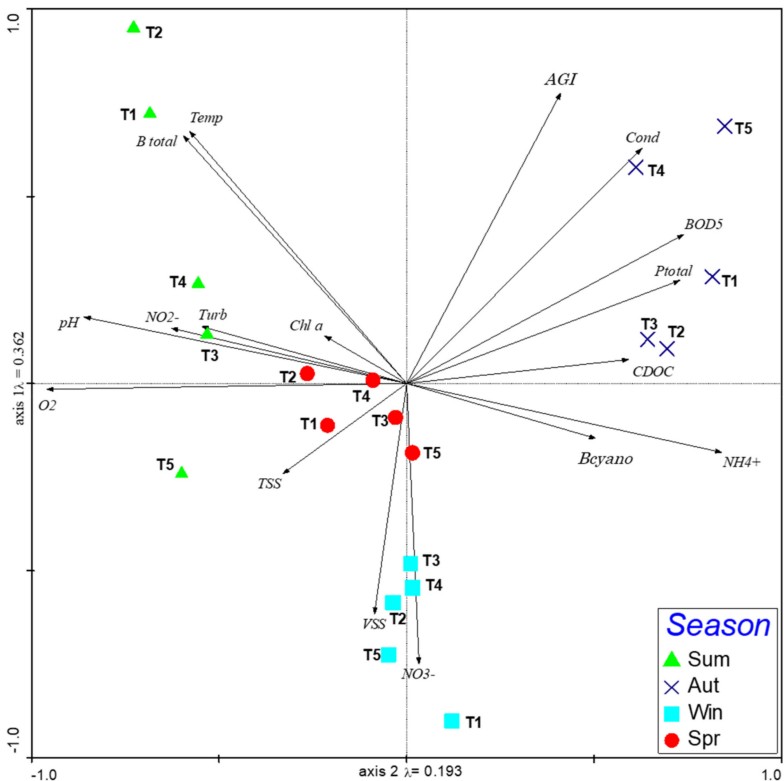

**Figure 2.** Spatial ordination resulting from the principal component analysis using the WFD parameters [*pH*, total phosphorus (*Ptotal*), nitrate ($NO_3^-$), dissolved oxygen ($O_2$), conductivity (*Cond*), temperature (*Temp*), nitrites ($NO_2^-$), ammonia ($NH_4^+$) biochemical oxygen demand ($BOD_5$), total of suspended solids (*TSS*), turbidity (*Turb*), dissolved organic carbon (*CDOC*), volatile suspended solids (*VSS*), chlorophyll *a* (*Chl a*), total biovolume (*Btotal*), cyanobacteria biovolume (*Bcyano*), and algae group index (*AGI*)]. Sampling sites: T1 to T5.

Table 2 also shows the values of the Shannon–Weaver diversity and Pielou evenness indexes for the zooplankton communities. The lowest values of diversity and evenness were recorded in spring for all sites. Sites T2, T3, and T4 showed the highest diversity values in summer, whereas in T1 and T5 this was observed in autumn. Regarding Pielou evenness values, T1, T4, and T5 presented the highest values in winter, while the remaining sampling sites showed the highest values in summer. Despite the differences recorded in the abundance and richness of *taxa* in each site, a clear seasonality was observed in this community, with the replacement and abundance of species occurring throughout the seasons. In spring, the sites were found to be quite similar (the same species and abundances), with Cyclopoid being the most responsible for this result since their relative abundance was above 80% (Table 2).

Large Cladocera/total Cladocera and Calanoida/Cyclopoida ratios showed the highest values in autumn and spring in site T1, while in the remaining sites, this was observed in summer and autumn (Table 2). These ratios were not calculated in the winter, since large Cladocera were not observed, except in T5. The Calanoida/Cyclopoida ratio presented the highest values in the summer and autumn. Calanoida was not observed in any of the sites in winter and in T5 in the spring.

**Table 2.** Results of the zooplankton community: abundances, diversity, and evenness indexes, and zooplankton ratios.

| Site | Season | Cyclopoida | Calanoida | *Chydorus* | *Pleuroxus* | *Alonella* | *Alona* | *Acroperus* | *Monospilus* | *Bosmina* | *Ceriodaphnia* | *Scapholeberis* | *Daphnia* | *Sida* | *Diaphanosoma* | Abundance | Diversity (Shannon—Wiener) | Evenness (Pielou) | Large Cladocera/Total Cladocera | Calanoida/Cyclopoida |
|---|---|---|---|---|---|---|---|---|---|---|---|---|---|---|---|---|---|---|---|---|
| T1 | Summer | 3611 | 159 | 1097 | | | | 14 | | 474 | 438 | 10 | 1 | 11 | 79 | 5884 | 1.19 | 0.54 | 4.30 | 0.04403 |
| | Autumn | 2884 | 404 | 877 | | | | 1 | | 56 | 2946 | | 47 | 125 | 601 | 7951 | 1.47 | 0.64 | 16.58 | 0.14008 |
| | Winter | 8 | | 17 | | | 61 | 5 | | 3 | | | | | | 94 | 1.07 | 0.66 | | |
| | Spring | 7395 | 1 | 2 | 1 | | 22 | 4 | | 494 | 4 | | 695 | | 1 | 8619 | 0.53 | 0.23 | 56.91 | 0.00014 |
| T2 | Summer | 373 | 167 | 287 | | | | 1 | | 32 | 66 | | | 127 | 175 | 1228 | 1.74 | 0.84 | 43.90 | 0.44772 |
| | Autumn | 314 | 226 | 279 | | | 1 | | | 3 | 531 | 9 | 4 | 29 | 68 | 1464 | 1.59 | 0.69 | 10.93 | 0.71975 |
| | Winter | 33 | | 2 | | | 23 | 1 | | 20 | | | | | | 79 | 1.22 | 0.76 | | |
| | Spring | 9521 | 1 | 4 | | | 16 | 6 | | 2495 | 2 | | 24 | | 2 | 12,071 | 0.54 | 0.25 | 1.02 | 0.00011 |
| T3 | Summer | 194 | 346 | 1005 | 2 | 210 | | 384 | 1 | 22 | 366 | | | 862 | 35 | 3427 | 1.84 | 0.77 | 31.07 | 1.78351 |
| | Autumn | 53 | 69 | 46 | | | 2 | | 1 | 1 | 54 | 3 | | 1421 | 84 | 1734 | 0.78 | 0.34 | 93.36 | 1.30189 |
| | Winter | 107 | | 5 | | | 4 | 5 | | 2 | | | | | | 118 | 0.41 | 0.29 | | |
| | Spring | 7284 | 2 | 20 | | | 26 | 5 | | 260 | 26 | | | 82 | 6 | 7711 | 0.28 | 0.13 | 20.71 | 0.00027 |
| T4 | Summer | 392 | 2126 | 309 | | | | 5 | 4 | 7 | 697 | | | 13 | 1139 | 3741 | 1.27 | 0.58 | 16.44 | 5.42347 |
| | Autumn | 90 | 109 | 4 | | 1 | 2 | | 1 | 10 | 2651 | | 4 | 1493 | 39 | 4470 | 0.97 | 0.40 | 37.51 | 1.21111 |
| | Winter | 40 | | 5 | | | 2 | 8 | | 29 | | | | | | 76 | 0.98 | 0.71 | | |
| | Spring | 2388 | 3 | 6 | | 2 | 20 | 8 | | 81 | 15 | 10 | 17 | 5 | 7 | 2560 | 0.37 | 0.15 | 15.98 | 0.00126 |
| T5 | Summer | 581 | 6159 | 535 | 1 | 10 | | 6 | | 9 | 3984 | | | 21 | 188 | 12,445 | 1.24 | 0.54 | 20.33 | 10.60069 |
| | Autumn | 88 | 87 | 1 | | | | | | 7 | 291 | | 7 | 136 | 105 | 656 | 1.50 | 0.72 | 37.84 | 0.98864 |
| | Winter | 27 | | 13 | | | 5 | 8 | | 4 | | | | 1 | | 50 | 1.19 | 0.74 | 4.35 | |
| | Spring | 3573 | | 26 | | 1 | 36 | 8 | | 35 | 19 | 7 | 9 | 2 | 5 | 3723 | 0.25 | 0.10 | 12.00 | |

Figure 3 shows the distribution of zooplankton according to the functional groups described in [41]. In general, high efficiency feed filters and macrofiltrators dominate the Torrão reservoir in all seasons. Indeed, the high efficiency filter feeders showed higher abundances in summer (>20%) and autumn (>10% and <60%), except for site T1, where the highest abundance of this functional group was recorded in winter (>80%). The macrofiltrators were observed in high abundances in almost all sites, namely in spring, with the exception of T1 in winter (only Cyclopoida were recorded, and with a low abundance, as shown in Table 2). The low efficiency feed filters were observed residually in most of the sites, being more noticeable in winter (T2, T4, and T5) and spring (T1 and T2), but almost always with less than 20% of relative abundance.

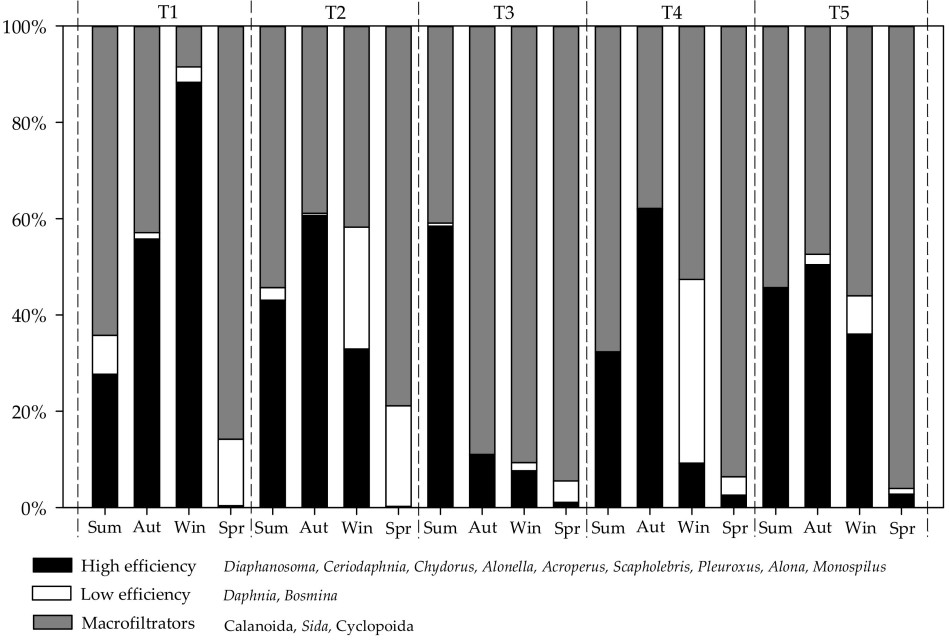

**Figure 3.** Relative abundance (%) of zooplankton functional groups for each sampling site (T1 to T5) in the four sampling periods (summer, autumn, winter, and spring) considering the filtering and feeding capacities [41,51].

In Figure 4, we analyzed the correlations between zooplankton and the physical, chemical, and biological parameters measured regarding the WFD approach. The abundance of zooplankton shows a significant positive correlation with temperature ($p = 3.4 \times 10^{-2}$) and a significant negative correlation with $NH_4$ ($p = 2.5 \times 10^{-2}$). Zooplankton diversity presents a significant positive correlation with total biovolume ($p = 2.6 \times 10^{-2}$), and zooplankton evenness has a significant negative correlation with chlorophyll *a* ($p = 4.1 \times 10^{-2}$). Zooplankton diversity and evenness indexes showed a significant negative correlation with Cyclopoida ($p = 1.1 \times 10^{-2}$ and $p = 2.5 \times 10^{-3}$, respectively) and macrofiltrators (such as Calanoida and *Sida*) ($p = 1.4 \times 10^{-4}$ and $p = 2.0 \times 10^{-5}$, respectively). The low efficiency organisms showed a significant positive correlation with the second ($p = 3.0 \times 10^{-2}$) and third cycle EQR ($p = 7.3 \times 10^{-3}$), which means that sites with better ecological potential present an increase in low efficiency organisms (such as *Daphnia* and *Bosmina*). The genus *Daphnia* showed a positive correlation ($p = 3.7 \times 10^{-2}$) with total suspended solids, which means an increase in the potential food available. Cyclopoida showed a significant negative correlation with parameters associated with eutrophic waters, such as ammonia ($p = 4.1 \times 10^{-2}$). The Calanoida/Cyclopoida ratio showed a significant positive correlation with temperature ($p = 2.4 \times 10^{-2}$); however, no relationship was observed regarding water quality.

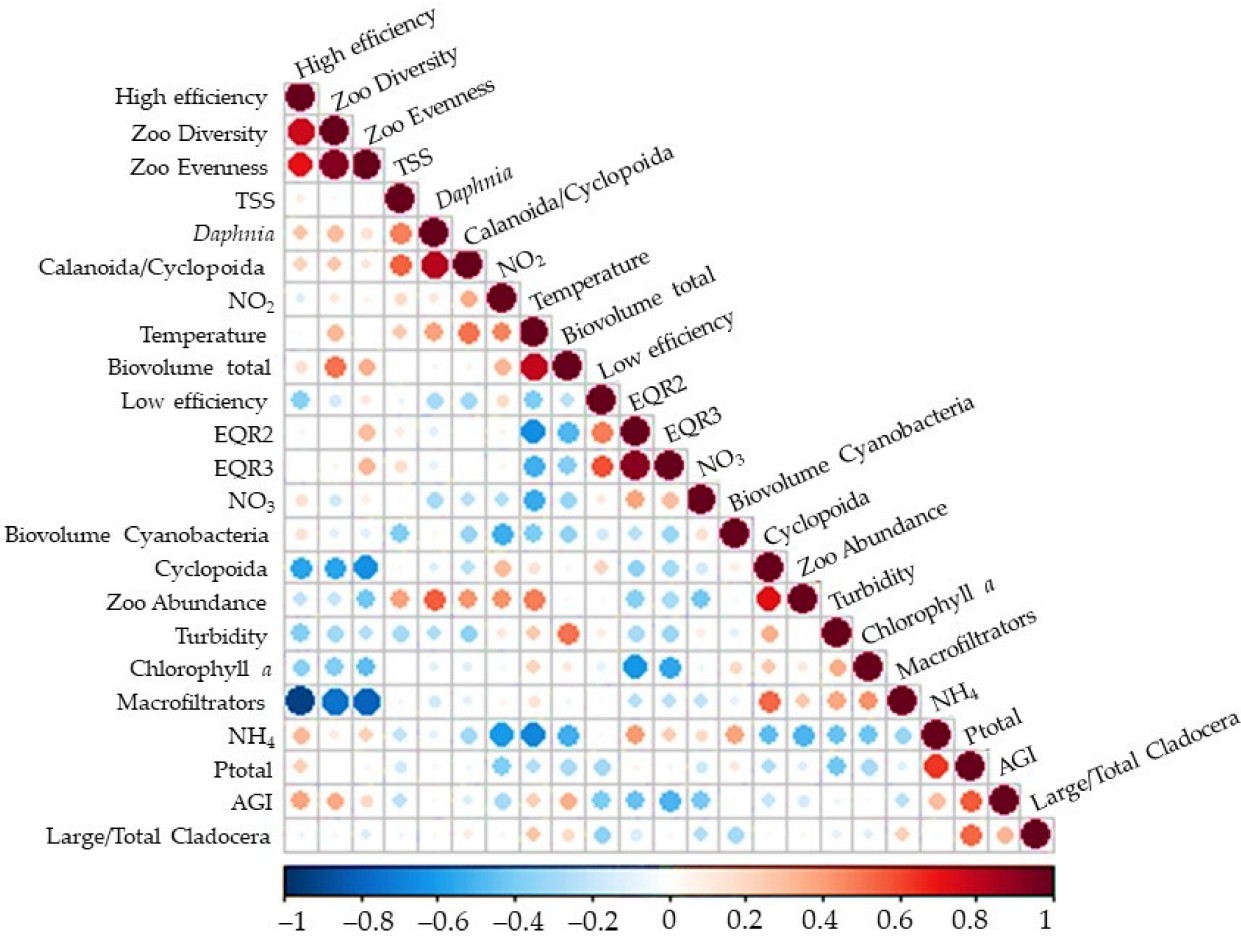

**Figure 4.** Results of the correlations between the zooplankton, physical, chemical, and biological parameters. Blue and orange gradients stand for negative and positive correlation, respectively.

## 4. Discussion

According to the official reports on the determination of ecological potential [52,53], the Torrão reservoir has been classified with a moderate ecological potential (2009 to 2021), with dissolved oxygen, pH, nitrite, and the phytoplanktonic elements being responsible for this classification. These results agree with those obtained in the present study, and it should be noted that the official assessment of this typology of reservoir is only based on summer sampling. In addition, total phosphorus and nitrate concentrations proved to be relevant parameters for the water quality achieved in this reservoir, being outside the reference values. Indeed, this reservoir shows an increase in nutrient concentrations during the last few years. Vale [54] observed high dissolved oxygen values in the summer of 2005, and Pereira et al. [55] observed high pH (>9) and dissolved oxygen values (>120%) in 2008. Furthermore, it should be noted that the sudden increase in phosphorus content in autumn may be related to the leaching of the terrestrial surrounding areas (first rains after a dry summer with numerous forest fires). The recognition of the existence of significant pressures, namely agriculture and wastewater (urban and livestock) [52] observed in the areas surrounding the sampling sites (Figure 1), may explain the lower water quality of this reservoir. Pinto et al. [12] reported that agriculture and untreated water discharges promote the accumulation of nutrients in water reservoirs. Moreover, Rodrigues et al. [3] and Bellinger et al. [56] also reported that a high load of nutrients (e.g., phosphorus), an increase in temperature, and adequate light conditions promote the rapid growth of the phytoplankton community (inducing water quality degradation).

Considering the variation in EQR, namely in terms of chlorophyll *a* concentration and cyanobacterial content (Table S2), it is possible to note that the reservoir is subject to

different pressures throughout the year. Pereira et al. [55] state that, in general, high pH values are associated with cyanobacterial dominance, with the occurrence of blooms, as we also found in our results (Tables S1 and S2). In many water basins of the Iberian Peninsula, phytoplankton has a spatial gradient along the reservoir, according to nutrient concentrations [57], and cyanobacteria are the main constituents of phytoplankton, representing an ecological risk for water bodies [58]. The fact that high biovolumes have been recorded that are associated with Cyanobacteria (*Nostoc*, *Microcystis*, and *Chroococcus*) and Chlorophytes (*Volvox*) (Table S2) could also be a reason for the lower ecological potential recorded in the Torrão reservoir. Indeed, studies carried out in this reservoir showed that cyanobacteria tend to present high densities in summer [54,59,60]. *Chroococcus* was observed in high abundance in the summer, contributing to the highest total biovolume recorded (Table S2). The high occurrence of *Chroococcus* can be related to the increase in water temperature and high pH (>8.5), which are ideal conditions for the proliferation of these organisms [56,59]. On the other hand, the presence of diatoms of the genera *Cyclotella*, *Synedra*, and *Nitzschia*, associated with cyanobacteria, are indicative of eutrophic environments [61]. These genera were recorded, in the present study, in almost all sites, with higher abundances in spring and summer, and identical results were recorded by [54] in the Torrão reservoir in 2003.

The revision of the WFD threshold values from the second to the third cycle further emphasized the lower ecological potential of this reservoir (Table 1). The increase in the number of physical and chemical parameters required in the third cycle, the existence of one more category (excellent), the lower threshold values, and the changes in the equation for calculating the normalization of the four components of the biological elements provides for a more realistic scenario. Moreover, these results allow a better identification of the most susceptible locations and, thus, more accurate recovery and management measures.

This reservoir is in a climate transition zone (see the description in Section 2.1: Study Area), so seasonality will always be present and evident in the dynamics of this ecosystem (water chemistry and biological communities). The zooplankton community responds easily to the physical and chemical conditions of the habitat and to changes in cascades of trophic events, such as phytoplankton blooms, which are inherent to the geographic location of the study area. These abiotic changes reflected seasonally (e.g., lowest temperatures in winter, Table S1) unavoidably affect the richness of its species, the densities (e.g., lower abundances in winter, Table 2), and promote changes in its diversity and evenness (e.g., Cyclopoida is highest in spring and lowest in winter). Since these organisms are phytoplankton predators (exercising a top-down control role, by feeding), we cannot neglect the role of this community in controlling and determining algal composition and abundance in the lentic ecosystems. As mentioned in other studies, and due to these characteristics, the zooplankton community was evaluated considering the ecosystem's geographic location, under the influence of seasonality, and this proved to be an adequate indicator of water quality [62,63].

Our study showed low abundances of zooplankton in winter, where all sites were classified with better water quality. Abrantes et al., [64], in the study carried out in the Vela lagoon (Quiaios, Figueira da Foz, Portugal), also recorded a low abundance of zooplankton in the winter months. Moreover, the results presented here can be explained by the decrease in the abundance of phytoplanktonic organisms (the decrease of [chlorophyll *a*] and total biovolume, Table S2) and consequently less food availability. This was observed in winter, where Copepoda was almost the only zooplankton observed, since cladocerans take advantage of resting stages in this period [65]. Hessen et al., [66] identified the trophic state, based on chlorophyll *a* content, and observed that this variable is the main factor for explaining the zooplankton diversity, since species richness increased. Haberman et al. [67] found that the Shannon-Wiener diversity index for the zooplankton community in eutrophic lakes tends to vary between 1.0–2.0, which, in general, is in line with our results (Table 2). Beaver et al. [29] reported that Cyclopoida and *Bosmina* are characterized by being tolerant to eutrophic ecosystems. In this study, the high abundance of Cyclopoida was recorded in spring, which showed a lower ecological quality at four of the five sites sampled.

Stamou et al. [62] reported that monitoring the Cladocera community is essential for water quality assessment, since alterations in different functional groups (e.g., daphnids, bosminids, and chydorids, using different feeding strategies [41]), reflect changes in other communities and provide critical information for the functioning of the food web [17]. Thus, large-bodied zooplankton communities are indicative of lower fish predation and increased grazing pressure on phytoplankton. In this situation, high zooplankton biomass is dominated by large-bodied species (e.g., daphnids), eventually leading to clearer water, a fact that was not observed in the present study. Most of the eutrophic temperate freshwater ecosystems exhibit prolonged cyanobacterial blooms [62]. In addition, Brito et al. [19] demonstrated that phytoplankton of low nutritional quality, combined with the toxicity of several species (e.g., *Microcystis*, also observed in the present work) were responsible for the collapse of the *Daphnia* population and a shift towards small-bodied species in eutrophic reservoirs. Another aspect is the 'top-down' process, where large zooplankton species are susceptible to visual fish predation [67], which is size-selective, and the measurement of size frequencies provides information on this. The presence of omnivorous fish species (e.g., *Cyprinus carpio, Rutilus rutilus, Alburnus alburnus* and *Gambusia holbrooki*) has already been described in the Torrão reservoir, and these are strongly associated with zooplankton predation, eutrophic status, and low ecological quality [68].

Brito et al. [19] state that differences in the abundance and diversity of Copepoda between reservoirs can also be explained by the trophic state. Calanoida is associated with more oligotrophic ecosystems (better water quality), while Cyclopoida is associated with eutrophic lakes and reservoirs (poorer quality) [31]. Our results, despite the ecological potential being poor to moderate, show that Cyclopoida only increased marginally, in spring, in all sites, while Calanoida increased marginally in the summer in T3, T4, and T5, and autumn in T1 and T2. Although these results are contradictory to those previously mentioned, other authors have obtained the same pattern (a Calanoida increase with worse water quality) in Spanish reservoirs (also in the Iberian Peninsula) [63]. Haberman et al. [67], in the second largest lake in the Baltic countries ($270 \text{ km}^2$), described that the Calanoida/Cyclopoida ratio is a good indicator of ecological quality, since with an increase of the trophic state, Calanoida organisms decrease and Cyclopoida increase, and the ratio decreases with the appearance of eutrophication. This fact is in agreement with our results, while this ratio showed a strong and positive association with the genus *Daphnia* (phytoplankton controller), as well as some negative associations with the trophic state.

Based on the present study, the Torrão reservoir water quality does not reach the good ecological potential intended by the WFD. In addition, it was possible to observe that, with the changes in the WFD criteria from the second to the third cycle, the classification of this waterbody tends to get worse. The zooplankton community proved to be an important biological element in reservoirs, since when the classification of the WFD was the same, the zooplankton was able to be more descriptive regarding the water quality. The use of metrics, based on the presence and abundance of *taxa*, can differentiate the water quality, and in this study, Cyclopoida and *Bosmina* were associated with the worst quality, while Calanoida and *Daphnia* were associated with better quality, facts already described by several authors, such as Beaver et al. [29], Stamou et al. [62], Muñoz-Colmenares et al., [63], and Brito et al. [19]. The Shannon–Wiener diversity index (specific ranges for each trophic state) is also described as an important parameter in water quality assessment, as demonstrated by Haberman et al., [67]. Functional groups, namely high and low efficiency feed filters and macrofiltrators, (Geller and Müller [41], and ratios such as large Cladocera/total Cladocera and Calanoida/Cyclopoida (higher values associated with better water quality) also provide valuable information about pressures, namely the trophic state and the cyanobacterial and fish communities that induce changes in the zooplankton community and, consequently, on water quality [67]. Despite not being included as a biological quality element in the WFD strategy for reservoirs, we strongly reinforce the use of zooplankton as a complement bioindicator already proven for a broader range of reservoirs by the previously cited authors. However, it is important to identify the disadvantages/advantages

of using these biological groups; the identification requires skilled personnel (and for the phytoplankton, as well), but the counts are easier and less error-prone, since we work with larger organisms, and the sample preparations are quicker and easier (phytoplankton needs weeks for sedimentation and decantation).

Zooplankton is considered an important element in the structure and function of aquatic ecosystems, with a fundamental position in aquatic food webs (they are the energy link between primary producers such as phytoplankton and higher consumers such as fish) [62]. Within the Iberian Peninsula, the use of zooplankton as bioindicators of trophic status in reservoirs has been evaluated and recommended in different basins such as the Ebro [30,69], Cavado [70], and Jucar [71]. This work is a case study of a small reservoir with temperate climatic characteristics. Despite that the results obtained are in accordance with several studies that showed the key role of zooplankton in lentic ecosystems and water quality. Notwithstanding this, further research including reservoirs from different geographical areas, with various trophic status and pressures, will be undertaken. Moreover, other metrics, such as those described by Gomes et al., [72] and Korponai et al. [73], and microzooplankton organisms, should also be evaluated in the future to understand if these communities are sensitive bioindicators to assess water quality (responsive to different reservoirs with different pressures and locations). This recognition will eventually lead to the inclusion of zooplankton in monitoring programs for this type of ecosystem.

**Supplementary Materials:** The following supporting information can be downloaded at https://www.mdpi.com/article/10.3390/w15091678/s1. Table S1: Results of physical and chemical parameters [total phosphorus ($P_{total}$), nitrate ($NO_3^-$), pH, dissolved oxygen ($O_2$), conductivity (Cond), temperature (Temp), nitrite ($NO_2^-$), ammonia ($NH_4$), biological oxygen demand ($BOD_5$), total suspended solids (TSS), turbidity (Turb), dissolved organic carbon (CDOC), and volatile suspended solids (VSS)] measured in the water samples and in comparison with the threshold values of the second and third Portuguese cycles of the WFD. The colors (blue, green, and yellow) represent the category that they fit according to the ecological quality standards. Table S2: List of genera and respective total biovolumes ($mg/m^3$) observed in each sampling (summer, autumn, winter, spring) and for each location (T1 to T5). Additionally, the results of each biological parameter analyzed (chlorophyll *a*, total biovolume, cyanobacteria biovolume, and Algae Group Index (AGI)) can also be found there.

**Author Contributions:** I.P, S.N., S.R., N.F. and S.C.A. participated in the research and/or article preparation and conceptualization. I.P., S.N., S.C.A. and N.F. carried out the field and laboratory work. I.P. and S.N. wrote the original draft, and all authors performed the final review. I.P, S.N., S.R., N.F. and S.C.A. All authors have read and agreed to the published version of the manuscript.

**Funding:** This work was supported by the strategic programs of the Portuguese Foundation for Science and Technology with the references—UIDB/04423/2020 and UIDP/04423/2020. Sara Rodrigues and Sara Antunes were hired through the Regulamento do Emprego Científico e Tecnológico—RJEC from the Portuguese Foundation for Science and Technology (FCT) program (2020.00464.CEECIND and CEECIND/01756/2017, respectively). Ivo Pinto was supported by a Ph.D. grant (ref. 2022.l0194.BD) from the Portuguese Foundation for Science and Technology.

**Institutional Review Board Statement:** Not applicable.

**Informed Consent Statement:** Not applicable.

**Data Availability Statement:** Not applicable.

**Conflicts of Interest:** The authors declare that they have no known competing financial interests or personal relationships that could have appeared to influence the work reported in this paper.

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
