# Peer review of "Can Zooplankton Add Value to Monitoring Water Quality? A Case Study of a Meso/Eutrophic Portuguese Reservoir"

_water, doi:10.3390/w15091678_

Round 1

Reviewer 1 Report

The authors presented that although zooplankton is not involved in WFD as a biological element, and concluded that zooplankton is a good descriptor of the ecological status of water bodies. They investigated some possible metrics of zooplankton (Calanoida/Cyclopodia abundance ratio, Large/Total Cladocera abundance ratio, and feeding trait). They found that these metrics can be used to establish the ecological status of water bodies. The selected feeding trait (low, highly efficient bacterial filterers and macrofiltrator) is inappropriate since planktonic cladocerans have different filtering apparatuses. Bosminids are selective filterers; they can filter small algae and bacteria more effectively, while the larger species (Daphnia, Diaphanosoma) filter feed in a consistent manner, grazing upon all sizes. Since the body length of zooplankton is an important functional trait, thus biomass ratio is a better predictor than the abundance ratio. I gave the relevant references.

All my comments can be found in the reviewed pdf.

Author Response

Please find in the attached file the responses to the reviewer

Reviewer 2 Report

This study provides seasonal information at multiple sites on a single Portuguese reservoir.  The field work is creditable, and the data presented in a generally acceptable way.

The purpose of the study is to support the inclusion of large zooplankton (calanoids and cladocerans) as a complementary metric in the WFD trophic assessment mechanism.  I think that such an assessment is just not possible with data from a single reservoir.  Broadly applicable tools for assessing trophic states need to be tested against a suitably wide range of lakes and reservoirs.  As it stands, this is a seasonal study of water chemistry, phytoplankton and large zooplankton in the Torrão reservoir. The correlation element of the data analysis I find difficult under this circumstance.  I fear that seasonality is dominating these correlations.

I think that we know that there are associations between trophic state and zooplankton composition.  The issue is to tease out whether this is sufficient to use as a technique for assessing trophic state that complements, and hopefully is addtional, to the existing variables in the WFD. I have to find that it is not well suited to addressing this aim.  

Author Response

(The authors gave the same response as above.)

Round 2

Reviewer 2 Report

This is my second review of this manuscript.  My primary concerns with the first version were (1) the framing of the study, which appeared to be an analysis of the value of adding zooplankton to the standard list of variables for monitoring water quality, and (2) an over-analysis of the results.

To quiet a large extend these concerns remain, though the authors have partially defused the first insofar as they now stress that this is intended as a case study.  I still think that some more thinking is required.

Firstly the title raises unmet expectations.  This could perhaps be changed to something like "Can zooplankton add value to monitoring protocols or lake water quality status: a case study of a mesotrophic Portuguese reservoir"

There is no doubt that zooplankton are impacted by water quality.  Often the reason for not including them in broad monitoring guides is the need for skilled personnel to identify and count, and the expense that the time spent on such counts entails.  I think that to better evaluate the use of zooplankton the paper would really benefit from discussion of the added value of  including zooplankton counts in terms of understanding the lake system.  Indeed on line 480 the authors state that this is a cost-effective method, without any detailed discussion of the cost and the effectiveness.

As it stands then the conclusion seems to be that in this case study various aspects of the zooplankton populations correlated with EQR and the zooplankton were consistent with expectations. What is missing is a discussion of what value the zooplankton added to understanding, and hence why it is of value and "cost effective".

In addition, I think that the data are still over interpreted.  I can see no significant changes in this respect from the first submission.  Essentially we are looking at four site in a single water body, and some difference between them can be expected because this is a linear reservoir and the ecosystem is, presumable, transitioning from fluvial to lacustrine.  And it is sampled through four seasons.  The statistical analysis seems to treat the samples (time x site) as fully independent, and that does not seem to be the case.  Figure 2 suggests that season dominates the water quality (WQ) variables, and given the seasonality of zooplankton then it is of no surprise that there are correlations with WQ variables. 

As I commented in my previous review, I do feel that the study is trying to do something that only a comparative approach can do.  

A minor point - It is worth stressing that this reservoir is close to a (northward-moving) climate transition zone in the study site section, but it would help to give a little more information about this.  Currently it is only mentioned in the discussion.  I am not sure though why this means that there will be seasonality?  ( line 400)

Author Response

Water ref: 2269513

The negligence of zooplankton as a biological key component in the evaluation of heavily modified water bodies quality

Reviewer 2

This is my second review of this manuscript.  My primary concerns with the first version were (1) the framing of the study, which appeared to be an analysis of the value of adding zooplankton to the standard list of variables for monitoring water quality, and (2) an over-analysis of the results.

To quiet a large extend these concerns remain, though the authors have partially defused the first insofar as they now stress that this is intended as a case study.  I still think that some more thinking is required.

R: We thank the reviewer’s comments and suggestions to improve the manuscript. In fact, our goal is present a case study, in a small reservoir with a pronounced seasonality (with abiotic and biotic changes), where water quality was assessed by the WFD endpoints and perceive if zooplankton community may contribute, as a complementary biological element, to relevant data to achieve the water quality classification. Zooplankton has been identified as an important bioindicator in water quality evaluation, by other authors, in large lakes and reservoirs [19,21,27-31]. So, we believe that our study is a relevant contribution to this approach, focused on a small reservoir with a marked seasonality. Indeed, the results here obtained showed that despite the reservoir characteristics the zooplankton community seems to be a promising biological element in the assessment of water quality (namely changes in diversity, evenness, Large Cladocera/Total Cladocera, Calanoida/Cyclopoida).

Firstly the title raises unmet expectations.  This could perhaps be changed to something like "Can zooplankton add value to monitoring protocols or lake water quality status: a case study of a mesotrophic Portuguese reservoir"

R: We appreciate the suggestion of the reviewer and the title was rephrased for: “Can zooplankton add value to monitoring water quality? A case study of a meso/eutrophic Portuguese reservoir.”.

There is no doubt that zooplankton are impacted by water quality.  Often the reason for not including them in broad monitoring guides is the need for skilled personnel to identify and count, and the expense that the time spent on such counts entails.  I think that to better evaluate the use of zooplankton the paper would really benefit from discussion of the added value of including zooplankton counts in terms of understanding the lake system.  Indeed on line 480 the authors state that this is a cost-effective method, without any detailed discussion of the cost and the effectiveness.

R: We agree with the reviewer and some information about the advantages of the zooplankton, as a complementary Biological Quality Element within the Water Framework Directive (in this type of ecosystem), was added.

As it stands then the conclusion seems to be that in this case study various aspects of the zooplankton populations correlated with EQR and the zooplankton were consistent with expectations. What is missing is a discussion of what value the zooplankton added to understanding, and hence why it is of value and "cost effective".

R: We agree with the reviewer and some information was added to the manuscript.

In addition, I think that the data are still over interpreted.  I can see no significant changes in this respect from the first submission.  Essentially we are looking at four site in a single water body, and some difference between them can be expected because this is a linear reservoir and the ecosystem is, presumable, transitioning from fluvial to lacustrine.  And it is sampled through four seasons.  The statistical analysis seems to treat the samples (time x site) as fully independent, and that does not seem to be the case.  Figure 2 suggests that season dominates the water quality (WQ) variables, and given the seasonality of zooplankton then it is of no surprise that there are correlations with WQ variables. 

R: We understand the point of view of the reviewer, but we intend to analyse all the data since it is a case study, being only working in a reservoir we try to understand if small changes within the same reservoir can be reflected in the zooplankton community. As the reviewer says, differences are expected when comparing the sites further upstream with those closest to the dam. Based on the literature (high nutrient concentrations, the occurrence of algal blooms as well as higher temperatures tend to occur in the sites more distant from the dam), we tried to understand if these small differences can be observed in this reservoir. We tried to focus on the main groups (Cyclopoida, Calanoida, Daphnia, and Bosmina) that according to the literature are associated with better or worst water quality, in addition to these we used specific metrics (indices and ratios) since they are also used in other works such as promising to achieve water quality of lentic ecosystems [29, 62-63, 67-68]. However, some parts of the discussion were rewritten in order to simplify this approach. Regarding the statistical analysis, we treat the samples as independent once again due to the seasonality variation that occurs in this geographical area. Indeed, these results emphasized the necessity to collect samples in all seasons and in different sites of the aquatic ecosystem to achieve a more realistic water quality classification.

As I commented in my previous review, I do feel that the study is trying to do something that only a comparative approach can do.  

A minor point – It is worth stressing that this reservoir is close to a (northward-moving) climate transition zone in the study site section, but it would help to give a little more information about this.  Currently it is only mentioned in the discussion.  I am not sure though why this means that there will be seasonality?  ( line 400)

R: We appreciate the reviewer’s comment. Effectively, the geographical location and Mediterranean and oceanic influences were now further described (e.g. Portugal's climate), so that seasonality is understood as something inherent to the geographical position. In this sense, more information has been included in section 2.1. Area of study, as well as in the discussion.

Round 3

Reviewer 2 Report

The authors have made significant changes to the manuscript and have added text that goes some way to addressing my concerns. Acknowledging weaknesses is a better way of dealing with them than ignoring. 

Please can there be one more sweep for English language corrections, then I think that this paper will be able to enter the literature.